# Multipurpose Prevention Technologies: Oral, Parenteral, and Vaginal Dosage Forms for Prevention of HIV/STIs and Unplanned Pregnancy

**DOI:** 10.3390/polym13152450

**Published:** 2021-07-26

**Authors:** Isabella C. Young, Soumya Rahima Benhabbour

**Affiliations:** 1Division of Pharmacoengineering and Molecular Pharmaceutics, UNC Eshelman School of Pharmacy, University of North Carolina at Chapel Hill, Chapel Hill, NC 27599, USA; iyoung4@live.unc.edu; 2Joint Department of Biomedical Engineering, North Carolina State University and The University of North Carolina at Chapel Hill, Chapel Hill, NC 27599, USA

**Keywords:** contraceptives, drug delivery, HIV, implants, injectables, intravaginal rings, multipurpose prevention technologies, sexually transmitted infections, sustained release

## Abstract

There is a high global prevalence of HIV, sexually transmitted infections (STIs), and unplanned pregnancies. Current preventative daily oral dosing regimens can be ineffective due to low patient adherence. Sustained release delivery systems in conjunction with multipurpose prevention technologies (MPTs) can reduce high rates of HIV/STIs and unplanned pregnancies in an all-in-one efficacious, acceptable, and easily accessible technology to allow for prolonged release of antivirals and contraceptives. The concept and development of MPTs have greatly progressed over the past decade and demonstrate efficacious technologies that are user-accepted with potentially high adherence. This review gives a comprehensive overview of the latest oral, parenteral, and vaginally delivered MPTs in development as well as drug delivery formulations with the potential to advance as an MPT, and implementation studies regarding MPT user acceptability and adherence. Furthermore, there is a focus on MPT intravaginal rings emphasizing injection molding and hot-melt extrusion manufacturing limitations and emerging fabrication advancements. Lastly, formulation development considerations and limitations are discussed, such as nonhormonal contraceptive considerations, challenges with achieving a stable coformulation of multiple drugs, achieving sustained and controlled drug release, limiting drug–drug interactions, and advancing past preclinical development stages. Despite the challenges in the MPT landscape, these technologies demonstrate the potential to bridge gaps in preventative sexual and reproductive health care.

## 1. Introduction

### 1.1. Prevalence of HIV, STIs, and Unplanned Pregnancy

HIV, sexually transmitted infections (STIs), and unplanned pregnancy are global health crises that affect millions of women and men worldwide. As of 2019, there are approximately 38 million people worldwide living with HIV [1], over 1 million new STI cases each day [2], and almost half (45%) of all pregnancies are unplanned [3]. More specifically, sub-Saharan Africa accounts for over two-thirds (25.7 million) of all global HIV infections [4], and sub-Saharan African women and girls account for 59% of all new HIV infections [5]. Ultimately, all AIDS-related illnesses have resulted in almost 33 million deaths worldwide [4]. Moreover, according to the World Health Organization (WHO), there are 376 million new STI cases annually [2], and the yearly Center for Disease Control and Prevention (CDC) STI surveillance report showed an increase in STIs for the sixth consecutive year (2015–2021), with nearly 2.5 million combined cases of chlamydia, gonorrhea, and syphilis as of 2019 [6]. Approximately 80% of sexually active individuals are likely to contract human papillomavirus (HPV) [7], which is responsible for over 30,000 cases of cancer each year, including almost all cases of cervical and anal cancer, 75% of vaginal cancer, 70% of oropharyngeal cancer, and 69% of vulvar cancer in the United States [8]. In addition to HIV/STI prevalence, there are 74 million cases of unplanned pregnancies in low and middle-income counties annually, leading to 25 million unsafe abortions and 47,000 maternal deaths each year [9,10]. Reduced or poor prenatal care is found to be a consequence of unplanned pregnancies, which have been found to contribute to 2.7 million neonatal deaths and 2.6 million stillbirths each year [11,12,13].

### 1.2. Discontinuation of Current Treatments

Although there are many effective therapeutics currently available for HIV, STIs, and contraception, most are given as a daily oral regimen. For example, HIV pre-exposure prophylaxis (PrEP) provides early treatment to prevent post transmission of HIV infection [14,15]. However, PrEP along with other orally delivered antiretroviral (ARV) therapies possess limitations including the need for chronic administration, the possibility of drug resistance, and low patient adherence [16]. This holds true for the only two FDA-approved oral therapies using emtricitabine and tenofovir as a combination antiretroviral therapy for HIV PrEP. The average adherence to ARVs is approximately 70% and is one of the main reasons for failure rates and drug-resistant viruses [17]. The HIV Prevention Trials Network (HPTN) conducted a study (HPTN 082) to assess the causes for lack of PrEP adherence and determine strategies to help women comply with the daily PrEP dosing regimen. The authors of the study concluded that lack of adherence was mostly influenced by negative HIV stigma and disclosure concerns, and can be mitigated with long-acting technologies, clinic-based discussions, and clubs/activities to normalize sexual behavior and PrEP usage [18,19]. Moreover, WHO conducted a study investigating the discontinuation of contraceptives in low and middle-income countries and found that 65% of women with an unplanned pregnancy either did not use contraception or used traditional methods (e.g., withdrawal or calendar-based methods) [9]. Additionally, 41% of women who used short-acting modern methods (e.g., pills and condoms) discontinued their use due to side effects and the facile capability of discontinuation [9]. Overall, poor adherence to HIV/STI and contraceptive regimens remains high due to side effects, inconvenient dosing schedules, poor access to products, cost, low education level [20,21], or poor familiarity with products [20,21], and/or negative stigma, ultimately resulting in poor product efficacy [22,23]. 

### 1.3. Sustained Release Systems and Multipurpose Prevention Technologies 

Sustained released delivery systems can be utilized to bridge gaps in adherence where efficacy is strictly dependent on user compliance. Sustained release delivery systems are designed to achieve prolonged drug release within its therapeutic window after single-dose administration. These systems can improve drug performance by increasing the duration of drug action, increase patient compliance by decreasing dosing frequency, and can aid in reducing adverse side effects. There are many marketed sustained-release products for contraception (e.g., intravaginal rings; NuvaRing^^®^^, intrauterine devices; Mirena^®^, solid implants; Nexplanon^®^, injectables; Depo-Provera^®^) that have been shown to increase patient adherence, and thus efficacy, due to their long-acting ability. Additionally, emerging technologies for HIV PrEP are in development to promote the sustained release of ARV therapy [18,24,25,26] and a long-acting injection administered every four weeks, Cabenuva (cabotegravir/rilpivirine), has recently been approved for HIV treatment. 

In conjunction with sustained-release delivery systems, multipurpose prevention technologies (MPTs) offer other opportunities to improve preventative therapies. MPTs can be defined as single-entity formulations, technologies, or strategies designed to address at least two sexual or reproductive health indications [23,27,28]. The development of a long-acting MPT would allow for the control and prevention of HIV, other STIs, and unplanned pregnancies in an all-in-one technology increasing patient compliance and acceptability. Furthermore, it allows for the opportunity to increase the demand for one product type to achieve uptake of the second, in other words, effective contraception might interest women to use other products that also target HIV and/or other STIs [29]. The need for MPTs can be further emphasized due to the epidemiological synergy between some STIs and HIV [30]. According to the CDC, those who have herpes, syphilis, or gonorrhea, are more likely to contract HIV as those STIs cause breaks in the lining of the genital tract, thus resulting as entry points for HIV infection. One example of this is the biological correlation between HIV and herpes simplex virus type 2 (HSV-2). Active HSV-2 infections consist of high concentrations of activated CD4-positive T cells in the genital area, which are target cells for HIV, and breaks in the mucosal layer cause HIV to infect the individual at risk [31]. HSV-2 is one of the most common STIs worldwide and is associated with a fivefold increased risk for HIV infection [31]. MPTs can provide a solution to increase protection and reduce high rates of these sexual and reproductive health indications due to the myriad of advantages of sustained-release delivery systems, the correlation between HIV and STIs, and the unmet need for the control and prevention of HIV/STIs and unplanned pregnancy. 

Currently, condoms are the only marketed MPT but can result in high failure rates due to poor user adherence, poor partner cooperation, and low acceptability [22]. Fortunately, the concept and development of MPTs have greatly expanded in the past decade and now encompasses many dosage forms, such as vaginal gels, films, tablets, implants, and intravaginal rings [23]. Important considerations for MPT development should be addressed, namely, (1) minimal to no drug–drug interactions or combined effects, (2) active ingredients should have minimal to no systemic or local side effects, (3) formulation should be easy to manufacture and administer with minimal discomfort, (4) facile removal or reversibility of the formulation in case of emergency or adverse side effects, and (5) end-user preferences of target populations [23,32,33]. Table 1 presents examples of current MPTs in clinical development. Appendix A includes MPTs in preclinical or earlier development.

Ultimately, there is an urgent need to develop effective, low-cost, and easily accessible MPTs for the control and prevention against HIV, other STIs, and unplanned pregnancies. MPTs can open the opportunity to address many health concerns and improve sexual and reproductive health worldwide. This review will cover the latest oral, parenteral, and vaginal MPTs in development, as well as current limitations, formulation development considerations, and implementation studies.

## 2. Method for Literature Search and Collection of Articles

The authors searched a number of electronic databases for journal articles regarding the preclinical and clinical development of MPTs, namely, PubMed, Google Scholar, US National Library of Medicine Clinical Trials (http://clinicaltrials.gov, accessed on 23 July 2021), and the Initiative for Multipurpose Prevention Technologies (IMPT) product database (https://mpts101.org/, accessed on 23 July 2021). Additionally, we searched conference abstract and presentation databases from the Conference of Retroviruses and Opportunistic Infections (CROI), International AIDS Society (IAS), HIV Research for Prevention (HIVR4P), and Long-Acting/Extended-Release Antiretroviral Research Resource Program (LEAP). Using these databases, the authors searched for oral, parenteral, and vaginal MPTs in development with an emphasis on next-generation IVRs. Keywords included multipurpose prevention technology, HIV, STI, contraception, pod intravaginal ring, segmented intravaginal ring, 3D-printed intravaginal rings, injectable, implant, controlled release, vaginal films/gels, and nonhormonal multipurpose prevention technology. All orally and parenterally delivered MPTs were selected. Novel topical/vaginal MPTs were selected, such as controlled-release vaginal films/gels/tablets and nonhormonal vaginal MPTs. Pod and segmented MPT IVRs were selected, as well as all 3D-printed IVRs, to emphasize advances in IVR manufacturing and next-generation IVRs. Table 1 was generated based on MPTs in clinical development, and Appendix A was generated based on MPTs in preclinical development found during our extensive literature search utilizing the databases abovementioned.

## 3. Orally Delivered MPTs

To our knowledge, there are not any published data on orally delivered MPTs in development. There are extended-release oral dosage forms being developed for HIV and STIs [17,24], but none is being applied as an MPT. Oral MPTs may not be advantageous due to challenges with poor patient compliance and challenges with incorporating multiple active pharmaceutical ingredients (APIs) in a single tablet/capsule, especially those with low solubility and/or poor gastrointestinal (GI) permeability. Other challenges regarding oral delivery systems are low residence time in the GI tract and limited material choices that do not degrade by the GI tract’s acidic pH [17]. 

Despite limitations with oral formulations, the Population Council, a major player in the MPT landscape, is developing the first oral MPT for prevention of HIV and unplanned pregnancy, which is currently in phase 3 clinical trials [35,50,51]. The oral MPT would incorporate the Truvada^®^ regimen (tenofovir disoproxyl fumarate and emtricitabine, 300 and 200 mg, respectively), as well as combined hormone therapy for contraception (levonorgestrel and ethinyl estradiol, 150 and 30 mcg, respectively) [51]. This oral regimen would be taken daily for 28 days (21 days of the dual-purpose pill containing both HIV PrEP and contraceptive, followed by 7 days of only HIV PrEP), similar to a typical oral contraceptive regimen [50,51]. However, no data have been published to support the progress of this formulation.

When considering the successful development of an oral MPT, extensive formulation development is needed to achieve prolonged and tunable drug release and the ability to coformulate multiple APIs with different physiochemical properties. For example, Kirtane et al. developed a long-acting oral delivery system for ARVs. The authors overcame challenges with low GI residence time and were able to achieve tunable drug release. This was accomplished by developing a device with an elastomeric core and six drug-loaded arms to be folded into a single capsule shell [17]. The six arms could be fabricated with different polymers to control drug release, account for different physiochemical properties, and promote the ability to load multiple drugs in a single structure. Kirtane et al. tested their formulation by encapsulating ARVs such as dolutegravir, cabotegravir, and rilpivirine in various polymer matrices to assess different release kinetics. Ultimately, they found that drug release can be easily tuned to achieve prolonged release and utilized with a once-weekly dosing schedule [17]. Although these drugs were not tested in combination within the formulation, this technology has shown sustained drug release of multiple drugs in an oral delivery system and has the potential to act as an MPT. 

Another example of a long-acting oral formulation is the delivery of the extremely potent ARV, islatravir (4′-ethynyl-2-fluoro-2′-deoxyadenosine, ISL), currently in phase 2 clinical trials for once-weekly or once-monthly dosing for HIV PrEP or treatment [24,52]. ISL has an exceptionally long intracellular half-life, compared with other nucleoside reverse transcriptase inhibitors (NRTIs), and retains significant potency against a broad range of clinically important drug-resistant isolates, including HIV strains containing the K65R reverse transcriptase mutation [53,54]. ISL has been shown to provide protection against rectal simian-HIV (SHIV) challenge in a rhesus macaque model when administered orally once weekly at doses as low as 0.1 mg/kg [55,56]. Furthermore, ISL has been investigated in combination with an investigational non-nucleoside reverse transcriptase inhibitor (NNTRI), MK-8507. Data from a dose-selection model for a future phase 2b study showed promise for HIV treatment by administering once-weekly doses of ISL (20 mg) combined with MK-8507 (100–400 mg) [57]. Based on the pharmacokinetic (PK) model, these doses would provide at least 90% efficacy and antiviral activity against common NRTI and NNRTI resistance-associated variants and robust viral load suppression as well as efficacy in the event of a late or missed dose [58]. Furthermore, ISL is approaching phase 3 clinical development for once-monthly HIV PrEP therapy. A once-monthly oral dose (60 mg) of ISL was selected by a PK/PD model and predicted to reach above a selected PK threshold (5X IC50 = 0.05 pmol/10^6^ PBMCs) and maintain sustained exposures in the event of a late or missed dose [56]. It is important to note that the success of these prolonged oral formulations is directly correlated to the drug’s extremely high potency, thus would not be achievable with other ARVs. Incorporating ISL as the ARV in an MPT could aid in reducing the dosing frequency and reduce the total amount of drug-loaded in the MPT formulation due to its high potency. 

Many formulation considerations need to be addressed when combining multiple APIs in a single capsule or tablet. Orally delivered APIs can present challenges with drug–drug interactions during the first-pass metabolism, solubility and absorption issues (e.g., low bioavailability), and excessively large size capsules/tablets due to high doses or high drug loading. Oral delivery systems for prolonged release should require the use of mucoadhesive polymers and present high gastric residence times [59] or incorporate APIs with incredibly high potency. However, the issue of patient compliance and adherence would likely still be prominent if an oral MPT were to be developed. Despite these limitations, if an oral MPT were to be developed successfully, this would give women, especially those who want to conceive in the near future, a short-acting preventative option [60]. 

## 4. Parenterally Delivered MPTs

Parenteral administration of long-acting prevention methods is an advantageous platform for MPT development. Currently, there are many promising solid implants and injectable drug delivery systems for HIV prevention in clinical and preclinical development [25,26,61,62,63,64,65], a long-acting injectable for HIV treatment was recently approved (Cabenuva), and there are marketed parenterally administered long-acting methods for contraception (e.g., solid implants; Nexplanon^®^, injectables; Depo-Provera^®^). These technologies and platforms can be utilized and adapted towards MPT development. 

Based on an extensive literature search, there are only three long-acting parenterally administered MPTs in development. A recent phase I clinical data demonstrated that ISL administered as a long-acting subdermal implant for HIV PrEP can last up to one year [52,61]. The development of the drug-loaded implant was tested with various bioerodible and nonbioerodible polymer systems, such as poly(lactic acid), poly(caprolactone), and poly(ethylene vinyl acetate). The implants are compatible with a wide range of molecules with varying physiochemical properties, including those with high aqueous solubility and amorphous phases, which are typically inapt as solid drug suspensions [61]. ISL eluting polymeric implants were projected to release for more than 6 months in rodents and nonhuman primates after a single subcutaneous administration and projected to release for up to one year in phase 1 clinical trial [61,66]. In phase 1 clinical trial, a single ISL eluting (48 mg, 52 mg, or 56 mg) or placebo implant was placed in participants with low-risk HIV infection for 12 weeks. Safety and PK were assessed throughout placement and 8 weeks post-removal [66]. The implants were well tolerated and ISL implants loaded with at least 52 mg achieved ISL-triphosphate concentrations above the PK threshold (5X IC50 = 0.05 pmol/10^6^ PBMCs) for 52 weeks. 

Based on the promising results with the ISL implants, an MPT is being developed by coformulating ISL and etonogestrel (ENG) as a solid implant [67]. This solid implant MPT is based on the Nexplanon^®^ technology and is projected to last an entire year for contraception and HIV PrEP [67]. Data from an in vitro model showed success in achieving target drug release of the APIs for over 30 days [67]. No data have been shown regarding drug–drug interactions, which is crucial for the successful development of the MPT, nor have there been any previous studies on coformulating ISL with another API. Additionally, no in vivo data have been published. Furthermore, this implant would need to be surgically inserted and removed, which may be unfavorable to end users and/or not be feasible in developing countries where medical resources are limited. 

Another example of a subcutaneously administered MPT in development is a biodegradable implant projected to last up to 12 months for HIV prevention and contraception [68]. Li et al. coformulated levonorgestrel (LNG), etonogestrel (ENG), tenofovir alafenamide (TAF), and ISL in poly(caprolactone) tubes based on a reservoir-style TAF implant [66,67,68]. The formulation could be administered as two implants in-line with a single trocar or as a single segmented implant that includes different drug formulations in each compartment [68,69]. In vitro release studies of the MPT formulations achieved zero-order and sustained release profiles. MPTs with ISL released for up to 12 months and MPTs with TAF released for up to 4 months. Target release and long-term API stability were driven by excipients, such as castor oil and sesame oil. The authors’ in vitro data demonstrated facile tunability of release rates based on the amount of excipient in the formulation. For example, the more sesame oil in their formulation correlated to faster drug release [68]. To achieve the long-acting release of MPTs for 12 months, the authors fabricated a 50:35:15 wt% of ISL:ENG:sesame oil and a 50:25:25 wt% of ISL:LNG:sesame oil [69]. The release kinetics of ISL with either contraceptive is comparable to the release of ISL formulated alone in the implant. In addition, Li et al. fabricated an implant with 50:35:15 wt% TAF:ENG:sesame oil and 50:35:15 wt% of TAF:LNG:sesame oil and observed zero-order release kinetics for up to four months. The release rates of TAF were altered when coformulated with either contraceptive compared to when formulated alone [69]. These data demonstrated that coformulation of ARVs and contraceptive hormones can influence the release rate of certain APIs, which can be used to modify the release rates of ARVs and/or contraceptives during coformulation. Li et al. were able to further tune the release kinetics by altering the polymer tube dimensions, such as surface area or wall thickness, and polymer material and characteristics, such as molecular weight and crystallinity [70]. However, because there are no published in vivo studies, it is not clear if this technology can achieve the target and sustained concentrations for all drugs in relevant animal models. Nevertheless, the group previously performed in vivo studies with the implant loaded with TAF alone and achieved target release kinetics and demonstrated easy removal of the implant [71], if needed. Overall, because the implant is biodegradable with facile removal if needed, long-acting zero-order release kinetics and the ability to coformulate different combinations of ARVs and contraceptives in a single formulation demonstrate the promising potential of this MPT and warrant future in vivo safety, PK, and efficacy studies. 

Lastly, an MPT microarray patch is currently in development for HIV PrEP and contraception [72,73]. Rein-Weston et al. proposed to fabricate a microarray patch incorporating a progestin-based hormone and cabotegravir delivered via microneedles, which consist of an array of micron-scale projections (<1 mm in height) assembled on a baseplate and applied to the skin for delivery [72]. The patch would be approximately 20–140 cm^2^ with an ideal wear time of 20 min with weekly or monthly self-administration to achieve target efficacy [72]. However, the average size of most commercial microarray and transdermal patches for other biomedical applications is between 10 and 30 cm^2^ [74]. Thus, the proposed size of the MPT microarray patch is quite large, especially if fabricated on the larger range, making it unrealistic. In an effort to increase drug loading and duration of drug action, McCrudden et al. used modeling to estimate a patch size of 375 cm^2^ to maintain similar drug exposure over 28 days in humans [75], which is unrealistically large and would likely reduce user acceptability. 

The solid implants and microarray patch technologies mentioned above are the only parenterally administered MPTs with published data to our knowledge. However, there are other long-acting systems that could be adapted for MPT development. For example, Kovarova et al. and Benhabbour et al. utilized a long-acting subcutaneously administered injectable delivery platform for ARVs. The authors utilized the Atrigel^®^ in situ forming implant (ISFI) technology consisting of poly(lactic-*co*-glycolic acid) (PLGA) and *N*-methyl-2-pyrrolidone (NMP) that forms a solid depot upon phase inversion in aqueous conditions (Figure 1). ISFIs have been shown to promote sustained and tunable release profiles [76,77,78,79,80] and have been incorporated in many marketed biomedical applications (e.g., Eligard^®^, Atridox^®^, Sandostatin^®^). 

Kovarova et al. investigated the capability of this technology as an ultra-long-acting delivery system for dolutegravir and achieved sustained drug release in nonhuman primates and in humanized mice for up to nine months [62]. After a single administration of the dolutegravir-ISFI, acute HIV replication was inhibited and protected against repeated high dose vaginal HIV challenges in relevant primary transmitted viruses [62]. Furthermore, the authors demonstrated the ability for the implant to be removed to terminate the treatment if required. If there are no adverse events, the removal of the formulation is not necessary since PLGA will degrade over time with no toxic by-products. 

Using the same ISFI technology, Benhabbour et al. demonstrated the capability of coformulating multiple ARVs within the injectable solution and the ability to tune the in vitro release profiles by altering the polymer (PLGA) to solvent (NMP) ratio [81]. The authors were able to individually load six ARVs (dolutegravir, darunavir, MK-2048, atazanavir, rilpivirine, and ritonavir) in the ISFI formulation and assessed drug plasma concentration in mice for 30 days. Drug plasma concentrations were maintained above their protein-adjusted IC90 (PA-IC90) value over the 30-day time course and plasma concentrations for dolutegravir and MK-2048 were 1–2 logs above their PA-IC90 for over 11 and 4 months, respectively [81]. Furthermore, Benhabbour et al. demonstrated the ability to coformulate three ARVs in a single ISFI at different drug loadings, and all three ARVs achieved sustained drug plasma concentrations over 90 days [81]. All drugs used in this study have varying physiochemical properties (LogP and pKa), thus suggesting this technology to be compatible with many APIs at different concentrations as a single ISFI injection.

Although this formulation has many strengths and elicits potential as an MPT, the solvent commonly used in this technology (NMP) elicits toxicity; thus, safety concerns can arise from using a product that is designed for chronic and repeated administration [82]. This problem is especially concerning since NMP effluxes out of the system during the phase inversion, potentially exceeding its permitted daily exposure limit (5.3 mg/day) [82]. Furthermore, since there is a lag time between injection and formation of the solid depot, the initial drug burst release may exceed beyond the therapeutic window and cause systemic toxicity. Additionally, there can be variability with in vivo drug release kinetics depending on the shape and size of the ISFI depot, which is partly influenced based on the route of administration (e.g., subcutaneously or intramuscularly). For example, Patel et al. observed a uniform and spherically shaped depot in vitro, whereas a flat disc was observed after subcutaneous administration in vivo [83]. From this, it was found that burst release was always higher in vivo than in vitro, which can be attributed to implant swelling in vivo due to the interstitial pressure and compressive forces from the surrounding tissue [83,84]. Ultimately, one needs to consider safety and reproducibility concerns when formulating MPTs with this platform. 

The parenterally administered MPTs mentioned in this section are mostly in the early development stages and require further testing to determine their efficacy, effectiveness, and acceptance as MPTs. As opposed to oral formulations, injectables and implants have a much wider technical landscape owing to their facile scalability and larger design space, allowing for personalized medicine and the ability to promote a long-acting and sustained release. However, there are limitations with parenterally administered systems. For example, long-acting injectable nanosuspensions for HIV PrEP [63,85] typically cannot accommodate more than one drug and administration is irreversible. These challenges are also present in Cabenuva (cabotegravir/rilpivirine), the first approved long-acting injectable for HIV treatment containing separate nanosuspensions of cabotegravir and rilpivirine. Furthermore, injectable formulations face the problem of a subtherapeutic tail. During the tail, efficacy is lost, and a window opens for the development of HIV/STI resistance when infection occurs in the presence of subeffective systemic concentrations of ARVs. To mitigate this, one could consider a removable formulation that results in a complete and rapid reduction of ARV from plasma. Additionally, nonremovable long-acting injectable formulations will likely require an oral lead-in to ensure tolerability of the APIs. This becomes a burden on manufacturing, decreases patient adherence, can increase the cost of the therapy since it will require an additional formulation of a daily pill for the first couple of weeks. Other challenges with parenteral administration of MPTs include the need to maintain an implant’s small size or a small injection volume, which can compromise optimal drug loading, and a lack of acceptability from users if invasive surgery or large needles are required for administration. Thus, parenterally administered MPTs could incorporate a biodegradable material to prevent surgical insertion/removal, compatible with a wide range of APIs, specifically ones that are potent, and that can accommodate multiple drugs. Furthermore, one should consider developing a technology that can be easily removed in case of adverse events as well as to eliminate the need for an oral lead-in and potentially reduce the PK tail if the drug is rapidly eliminated once the implant is removed or fully degraded. 

## 5. Vaginally Delivered MPTs 

Since a vaginal MPT review was recently published [21], the scope of this section will only briefly discuss innovative vaginally delivered systems with an emphasis on intravaginal rings (IVRs). More information can be found in recent vaginal MPT review papers [23,28,86]. 

### 5.1. Vaginal Films, Gels, and Tablets

Vaginally delivered dosage forms include gels, creams, films, foams, suspensions, suppositories, and tablets. Current applications of these dosage forms are typically for gynecological maintenance, contraception, vaginal moisturizer or lubrication, or treatment for vaginal infections [87]. The effectiveness of vaginal delivery systems and microbicides is dependent upon the product’s bioadhesion, retention time, bioavailability, and user adherence [88]. 

Over the past decade, many vaginal dosage forms have been in development for HIV/STI prevention including few formulations as an MPT. One of the first promising vaginal regimens for HIV PrEP is the 1% tenofovir vaginal gel [42]. A phase 2b clinical trial (CAPRISA 004) was conducted to study its effectiveness and safety for HIV prevention in women. The gel formulation reduced HIV acquisition by approximately 39% overall, and by 54% in women with high gel adherence [42]. Later, the tenofovir gel demonstrated the ability to act as an MPT by preventing HSV-2 acquisition among women by 51% as demonstrated by the CAPRISA 004 trial and VOICE trial [89,90]. Another promising MPT vaginal gel in development contains MIV-150 and zinc acetate dihydrate in carrageenan to prevent HIV, HSV, and HPV and is currently in phase 1 clinical trials by the Population Council [36]. The microbicides gel showed a safe vaginal profile and has been shown to inhibit SHIV-reverse transcriptase in a macaque vaginal and rectal mucosal model and demonstrated antiviral activity against HSV-2 and HPV in murine models [36]. Currently, this is the only MPT product in clinical testing that promotes the prevention of HIV in addition to two other noncurable STIs.

Other dosage forms, such as vaginal tablets, films, and nanosystems, are in development that incorporate novel techniques as an attempt to control drug release. McConville et al. developed a multilayer vaginal tablet for contraception and prevention of HIV and HSV-2 with immediate and sustained drug release [91]. The tablet contains levonorgestrel (LNG), dapivirine (DPV), and acyclovir with the ability to control drug release rates by altering the number of tablet layers and the drug dosage. For instance, the tablet can promote the immediate release of all three drugs or provide immediate release of LNG and acyclovir with sustained release (up to eight hours) of DPV [91]. All release studies were performed in vitro; thus, it is difficult to determine how the system will interact with the vaginal environment in vivo. Moreover, although the authors were able to control drug release, a daily dosing schedule is still required. 

Similarly, Li et al. utilized LNG and DPV to develop an MPT film for contraception and HIV prevention [92]. A DPV vaginal film is currently in phase 1 clinical development; however, it does not exhibit significant mucoadhesiveness and controlled release [93]. Thus, Li et al. advanced the vaginal film formulation to include a contraceptive (LNG) and incorporated thiolated chitosan in the polyvinyl alcohol-based film to promote mucoadhesion and prolonged drug release. Thiolated polymers are known to bind strongly to mucins via covalent bond formation and have demonstrated in vivo tolerability, controlled release, and mucoadhesion at acidic pH (3.8–4.5), similar to the vaginal environment [94,95,96]. Li et al. characterized their DPV/LNG films, assessed ex vivo tissue toxicity, mucoadhesive properties, in vitro drug release, and in vivo PK and safety in macaques [92]. The mucoadhesive films were well tolerated in vivo, exhibited a higher retention time in macaques compared to films without thiolated chitosan, and DPV and LNG plasma levels were above their target threshold (7.9 ng/mL for DPV and 0.3 ng/mL for LNG) for up to seven days [92]. However, LNG plasma levels significantly increased when coformulated with DPV compared to an LNG single-entity film, potentially due to drug–drug interactions. Nevertheless, this MPT film has the potential to advance to clinical stages as a once-weekly film for protection and further studies evaluating in vivo efficacy are necessary. 

Most vaginal MPT dosage forms, including the ones mentioned above, utilize hormonal contraceptives. Although hormonal steroids are highly effective, some women experience side effects that lead to discontinuation and low user acceptability to the dosage form. Due to this, there have been attempts to develop nonhormonal vaginal MPTs. For example, Weitzel et al. developed a nonhormonal MPT vaginal gel utilizing polyphenylene carboxymethylene (PPCM) [97]. PPCM is an anionic mandelic acid condensation polymer that has been demonstrated to be active against HIV-1, HSV-1, and HSV-2 [98,99,100]. PPCM has also been shown to act as a noncytotoxic contraceptive that causes a premature loss of the sperm acrosome and does not cause epithelial surface damage unlike nonoxynol-9 [101]. The MPT formulation consisted of 4% PPCM sodium salt and excipients such as HPMC K100, xanthan gum, glycerin, methyl paraben, and propyl paraben and would be administered precoitally [97]. Only preliminary in vitro studies have been conducted to determine the contraceptive efficacy of this formulation and showed up to 80% sperm inactivation. However, this formulation is very early in preclinical development and further in vitro and in vivo testing is required to determine the safety, efficacy, and acceptability of this formulation. 

Ball et al. developed an innovative nonhormonal topical nanosystem for contraception and HIV prevention [102]. The authors utilized an electrospraying technique to formulate drug-loaded nanofiber meshes containing 1% zidovudine and 1% maraviroc for anti-HIV activity, and 10% glycerol monolaurate targeted against sperm motility. The authors investigated the drug-loaded formulation with a variety of polymers: poly(L-lactide), poly(ethylene oxide), polycaprolactone, and poly(d-lactide). Different polymer blends played a role in tuning in vitro drug release kinetics [102]. Ball et al. were able to achieve drug release up to six days with a 70:30 poly(l-lactide):poly(ethylene oxide) formulation and showed to inhibit HIV in vitro TZM-bL cell-based assay [102]. Although the authors demonstrated the inactivation of sperm motility with their proposed molecule for contraception, they did not show its effectiveness within the gel or when combined with the proposed ARVs or investigated any potential drug–drug interactions. Additionally, the mechanism of action of glycerol monolaurate on sperm motility is not yet known; thus, further studies are required to better understand this molecule as well as when combined with ARVs or other APIs. 

Another nonhormonal vaginal MPT is in development by Mapp Pharmaceuticals and is a polyvinyl alcohol-based vaginal film for HIV and HSV-2 prevention using monoclonal antibody product, MB66 [44,45]. This would be the first reported phase 1 clinical study of an antibody-based MPT for HIV and HSV-2 prevention. MB66 has shown to be efficacious against HIV and HSV-2 and acquires high specificity for target pathogens [44,45]. Politch et al. assessed repeated use of the film every 24 h for 7 days and achieved target drug release, successful viral neutralization, and the film was safe and well tolerated [44,45]. However, in this phase 1 clinical study, there is variability in film dissolution rate, which will influence changes in release kinetics, and PK testing was not conducted between days 1 through 7 [45]; therefore, further testing is required for this formulation. Nevertheless, due to the promising inhibitory effects of MB66, Anderson et al. suggested utilizing “human contraceptive antibody” (HCA) for an MPT vaginal film for HIV, HSV, and contraception [103]. HCA is the only antisperm antibody in advanced development and shown to promote sperm agglutination, trap flagellating sperm in human midcycle cervical mucus (mucus trapping), and promote viral neutralization of HIV and HSV [103]. HCA has been prepared for phase 1 clinical trials in a polyvinyl alcohol film containing maltitol, histidine, and polysorbate 20 with 10 mg of HCA to act as an MPT for contraception, and HIV and HSV prevention [103]. Although few published data are currently available for this product, there is potential for this technology to be a successful and novel MPT formulation. However, there may be challenges in large-scale antibody manufacturing and might not be as cost effective, compared to formulations with small molecule APIs. 

Although there are many effective vaginal formulations for a variety of applications, including MPTs, their efficacy is highly dependent on user adherence. Additionally, there are limitations with leakage, discomfort, and low mucoadhesive properties that need to be considered for these types of vaginal formulations to ensure efficacy and comfort to increase user acceptability and compliance. Vaginal dosage forms are also challenging to formulate due to the low and changing pH of the vagina and the abundance of proteolytic enzymes present in the genital tract that could potentially degrade the APIs or the material of the delivery vehicle. It is also important to ensure that the active and inactive ingredients do not interfere with the vaginal microbiome or mucosal epithelium. Ultimately, these formulations should consider approaches for tuning and prolonging drug release that incorporate mucoadhesive materials to enhance bioadhesion to the vagina and do not degrade at a low pH or alter the healthy vaginal microbiome. 

#### 5.2.1. Intravaginal Rings

Intravaginal (IVRs) are torus-shaped polymeric rings loaded with one or multiple APIs for controlled delivery via the vaginal tract. IVRs are advantageous drug delivery systems because they capitalize on the highly vascularized tissue that facilitates drug uptake and avoids the first-pass metabolism; they are also easy to use, long acting, and are not coitally dependent [22,104,105,106,107,108,109,110,111,112]. Moreover, IVRs are established as contraceptive devices [113,114,115,116], are amenable to a wide range of applications, compatible with many APIs, are women controlled, and are an appealing tool for MPT development. 

#### 5.2.2. Manufacturing Limitations and Emerging IVR Technologies

Currently, there are six marketed IVRs for contraception (NuvaRing^®^, Progering^®^, Fertiring^®^, Annovera^®^) and hormone replacement therapy (Femring^®^ and Estring^®^) that promote sustained drug release. A dapivirine microbicide IVR is close to registration for HIV PrEP [110,117], and several MPT IVRs are in development. Although IVRs are advantageous drug delivery systems, there are limitations in the manufacturing process that cause restrictions in IVR design and performance. Marketed IVRs and many of those in development are manufactured by hot-melt extrusion or injection molding. While traditional IVR manufacturing (i.e., injection molding and hot-melt extrusion) is fast and high throughput, it is limited to a solid cross-sectional design, and with the exception of pod IVRs, restricts separation between ring fabrication and drug incorporation, which involves high temperature and pressures limiting compatibility with API and material choices. Since achieving 100% drug release and controlled release is challenging due to the ring’s inevitable solid cross section, manufacturers often overload the ring with API in an attempt to achieve therapeutic efficacy [118]. This will increase API costs, which becomes a critical issue when the API is expensive. These limitations generally constrain IVR effectiveness, distribution, and potential for epidemiological impact. Various emerging techniques are attempting to mitigate these consequences from traditional manufacturing as well as incorporate multiple APIs with a range of physiochemical properties to achieve sustained and targeted drug release kinetics. Techniques include multisegment IVRs, incorporating tablet inserts or “pods” in the cavity of IVR, and 3D-printed IVRs (Figure 2). This review will focus on these emerging technological advances in IVR manufacturing and MPT development. MPT IVRs that are fabricated with standard injection molding techniques have been discussed in another review [23].

Segmented IVRs incorporate multiple polymer segments with different lipophilicities to account for API solubility differences, thus optimizing stability and release rates [38,105,119]. Clark et al. and CONRAD engineered a segmented MPT IVR to deliver tenofovir and LNG for 90 days for HIV prevention and contraception [38]. Since tenofovir and LNG acquire major differences in their partition coefficients and aqueous solubilities, the authors utilized chemically diverse polymers to effectively solubilize the APIs and allow for controlled delivery. A hydrophilic polyurethane was used to deliver tenofovir and a hydrophobic polyurethane to deliver LNG [38]. Clark et al. performed in silico, in vitro, and in vivo methodologies to assess drug release profiles and PK. The authors only assessed LNG concentrations in plasma during their PK studies in rabbits; thus, it is difficult to conclude how the dual–drug IVR would interact in vivo. Target in vitro and in vivo release rates were achieved for both APIs and levonorgestrel only, respectively. Furthermore, the group designed endcaps to prevent diffusion of the APIs into neighboring compartments, a major limitation faced with segmented IVRs [38,104]. Through mathematical simulations, the authors determined that after 2 years of storage, 6% LNG would have diffused into other compartments with the proposed endcap design [38]. Ultimately, segmented IVRs can load and deliver multiple APIs and offer more control over drug release, compared to traditional IVR manufacturing processes. However, less than 50% of the total drug load was released in vitro [38], suggesting that half of the drug load will not be utilized, which decreases the IVR cost efficiency. Additionally, unless rigorously designed, this approach can cause APIs to diffuse into neighboring segments and involves a strict and multistep manufacturing protocol while still incorporating traditional manufacturing techniques (e.g., hot-melt extrusion) to fabricate the segments.

Another evolving technology is the pod IVR, which is manufactured by incorporating API pellets into the IVR in individual pods created across the ring cross section [120,121,122,123,124,125]. Pod IVRs have been shown to promote controlled drug release by a rate-controlling membrane around the drug pellet and can easily incorporate multiple APIs [104]. Oak Crest Institute of Science and Marc Baum are major players in the pod IVR design space for HIV PrEP and MPT applications [122,123,124,125,126]. For example, Baum et al. fabricated a silicone pod IVR with tenofovir and acyclovir pellets coated with poly(lactic acid) for HIV and HSV-2 prevention [122,123]. In vitro and in vivo (rabbit and sheep) release studies, PK studies, and mechanical testing of the IVRs were performed. The authors demonstrated sustained zero-order release profiles of tenofovir and acyclovir for 28 days. However, only 60% and 15% of the total drug load was released during rabbit and sheep studies, respectively [123]. Overall, the authors’ pod IVR design elicited the ability to insert up to 10 pods of up to 40 mg of drug per pod, thus potentially able to release 400 mg of one API or 40 mg of 10 different APIs in a single IVR. Since this design was successful, Moss et al. incorporated five APIs within this pod IVR design and assessed PK and drug release in a sheep model [124]. The five APIs used in this model consisted of three ARV agents (tenofovir, nevirapine, and saquinavir) and two contraceptives (etonogestrel and estradiol). The IVR contained two pods of each API per ring, totaling 10 pods per ring with 16 mg of API per pod. In vivo sheep PK and release data showed that this pod IVR could deliver up to five APIs at effective concentrations and release rates required to potentially prevent HIV infection [124]. In the future, the authors would need to assess drug release kinetics, PK, and efficacy in a nonhuman primate model. Taking it further, Smith et al. used this technology to expand its indication for the prevention of HIV, HSV-2, and unplanned pregnancy [126]. Tenofovir, acyclovir, etonogestrel, and ethinyl estradiol were used in the IVR formulation, and in vitro and in vivo release kinetics were assessed for 30 days. Rigorous PK testing of the multidrug-loaded IVR was performed in macaques, and the authors assessed the distribution of each drug in plasma, vaginal tissue, and vaginal fluid [126]. Furthermore, Smith et al. demonstrated the ability to tune the release rate of etonogestrel by varying its drug loading [126]. However, similar to previous pod IVR formulations, less than 50% of the total drug load was released in vivo. Nevertheless, pod IVRs elicit many facets to promoting control drug delivery (e.g., pod polymer membrane, number of pods, cross-sectional diameter) and promote target PK and release profiles [123,124]. 

Ultimately, these two emerging technologies have shown the ability to load multiple drugs with different physiochemical properties in a single IVR, demonstrating the potential for successful MPT development and improvements in IVR fabrication. However, there is incomplete drug release (<50% of total drug load) due to the solid cross section constraining drug diffusion and resulting in overloading the ring with API [38,119,122,123]. Moreover, these IVRs still elicit IVR design restrictions and involves multistep, expensive, and time-consuming processes.

Additive manufacturing (3D printing) offers an opportunity to further improve upon IVR fabrication by expanding IVR design scope, thus allowing for greater control on drug release kinetics and the ability to achieve complete drug release from the IVR. To our knowledge, all MPT IVRs in development incorporate standard injection molding techniques to some extent, and there are only three published attempts to 3D print IVRs [127,128,129] based on an extensive literature search. For instance, Fu et al. demonstrated one of the first attempts to 3D print an IVR for the contraceptive delivery of progesterone [127]. A mixture of progesterone and polyethylene glycol was combined with poly(lactic acid), polycaprolactone, and Tween 80 and processed using hot-melt extrusion to produce polymer/drug filaments. Fused deposition modeling (FDM) 3D printing was used to generate “O”, “Y”, or “M”-shaped vaginal implants with the filaments [127]. Although the authors demonstrated that progesterone did not degrade or decompose during the hot-melt extrusion process, this may not be compatible with other APIs due to the applied high heat, a common limitation with injection molding. The IVRs were able to achieve in vitro sustained release of progesterone (100–200 µg/day) for over seven days and were projected to release progesterone above target concentrations for contraception [127]. 

Similarly, Welsh et al. used droplet deposition modeling (DDM) to 3D print a dapivirine-releasing IVR for HIV prevention. FDM and DDM are similar in that both techniques involve extruding a formulation through a nozzle to generate 3D structures from a computer-aided design (CAD) file. However, FDM involves a continuous extrusion of material, whereas DDM produces discrete streams of material during deposition [128]. Welsh et al. demonstrated the ability to increase IVR surface area by modulating the in-fill density and thus able to increase the amount of dapivirine released from the IVR by utilizing DDM [128]. The authors fabricated DPV loaded IVRs with DDM with different in-fill densities and compared the release kinetics to rings fabricated via injection molding. Welsh et al. found that those with low in-fill densities (10%) exhibited up to a sevenfold increase in drug release rate (in vitro), compared to injection molded rings (100% in-fill density) after 29 days. More specifically, the in vitro cumulative release of DPV over 29 days was up to 10% for rings with 100% in-fill density, 56% for rings with 50% in-fill density, and 79% for rings with 10% in-fill density [128]. These results demonstrate that CAD and 3D printing can be utilized to modulate the surface area of the IVR to allow for faster release kinetics, greater cumulative release of API, and greater tunability of release rates. However, both FDM and DDM require high heat and pressure, similar to injection molding, which could have deficits on ring mechanical properties as well as limit material and API choices. 

More recently, Janusziewicz et al. reported on 3D-printed IVRs fabricated using digital light synthesis or continuous liquid interface production (CLIP™) [129,130]. This method utilizes a photosensitive resin that selectively solidifies upon exposure to ultra-violet (UV) light via free radical photopolymerization mechanisms to generate the final product [130]. Uniquely, CLIP introduces oxygen into the system, which is known to inhibit the solidification or polymerization process by forming a region of unreacted resin called the dead zone [130]. This allows for continuous production of monolithic parts with smooth, nonlayered, and high-resolution structures at print rates upwards of 100 mm/hr. Janusziewicz et al. developed a design library of geometrically complex biocompatible silicone polyurethane-based IVRs, compared their mechanical properties to commercial rings (NuvaRing^®^ and Estring^®^), and assessed the fidelity of the CLIP-fabricated IVR to the original CAD file [129]. Incorporating geometric complexity to IVRs eliminates its solid crosssection and has the potential to promote targeted release kinetics and complete drug release as drug diffusion distance can be controlled. However, since there are no published data on drug-loaded CLIP-fabricated IVRs, it is difficult to determine if this system will achieve the target and controlled release kinetics. Nevertheless, Bloomquist et al. successfully incorporated APIs in photosensitive resins and demonstrated controlled drug release from CLIP-printed geometrically complex scaffolds [131]. From this, CLIP shows the ability for facile manufacturing and potential controlled and complete drug release of APIs due to the ability to vary drug diffusion distance by generating IVRs with geometrically complex internal architectures, which can eliminate the need to overload IVRs with APIs. Although CLIP has many advantages for developing 3D-printed IVRs, there are still limitations. For example, after printing, the silicone polyurethane IVRs require an 8 h and 120 °C postthermal cure step [129], which can degrade APIs that are sensitive to high heat if the drug is incorporated in the resin during printing. Additionally, UV exposure is a critical component of the CLIP IVR fabrication process, which limits API options that are sensitive to UV light if incorporating the drug into the resin. 

Ultimately, technological advances in IVR fabrication have been shown to further enhance the already established and valuable delivery system. However, there is room for improvement to achieve controlled and complete drug release kinetics without the need to overload the ring with API and the ability to load multiple APIs with different physiochemical properties in a single IVR. These attributes are essential for MPT development and have the potential to be achieved by altering the manufacturing process and expanding the IVR design space. 

## 6. MPT Implementation Studies 

Successful MPT development depends on technical formulation approaches and on social, economic, and behavioral aspects relating to adherence and acceptability of the product, particularly in developing countries and populations where HIV, STIs, and unplanned pregnancies rates are highest. Although there are many MPTs in development and some have shown efficacious results, products will only be effective in reducing HIV, STIs, and unplanned pregnancy rates if they are acceptable to end users. Gathering user input data will maximize the chances that technologies will move forward into efficacy trials and become available preventative options that are accepted and used correctly by women [132]. Many implementation studies have been conducted to demonstrate the acceptability and adherence of different MPTs. For example, the TRIO study examined the acceptability of three placebo MPT delivery forms: daily oral tablets, two monthly injections, and a monthly IVR [132,133,134]. The design of the implementation study is shown in Figure 3a. The authors conducted a randomized placebo crossover study that involved HIV-negative women in Kenya and South Africa to use each dosage form for a month followed by a usage period of two months with their dosage form of choice. Additionally, health care provider and male partner inputs were screened before and after the trial period, respectively [134]. Weinrib et al. considered four influences in acceptability (sociodemographic factors, social context, risk perception, and product features [132,135]) and examined product approval rating and product choice as implementation outcomes. From an initial TRIO study, most women preferred injections, followed by tablets, and lastly IVRs (Figure 3b). This was largely influenced by age, the importance of end users to dosing frequency, user burden, ease of use, and interference with daily activities [132,133,134]. Interestingly, almost all women who preferred rings at the end of the crossover period had favored a different product at the beginning of the study, thus emphasizing the importance for women to become familiar and educated with the various products [134]. This is especially important since IVRs are the most developed and advanced MPT option; thus, further emphasis should be given to improve its future acceptability; this can be accomplished by performing multiple focus groups with women at multiple age groups and different demographics and social status. Similarly, Van der Straten et al. conducted a similar TRIO study and concluded that injections were favored by women in South Africa and demonstrated full compliance and adherence toward injections, compared to IVRs and oral tablets (Figure 3c) [136]. Women in South Africa may prefer injections over other products because they are more familiar with parenteral administration, and there is less concern of it interfering with sex or daily activities [134,136,137]. It is important to emphasize that the TRIO studies were conducted with placebo dosage forms; thus, side effects and possible adverse events were not considered a factor in the study. Overall, the TRIO studies concluded that women are accepting towards MPTs with injections as their preferred MPT dosage form. 

In addition to user acceptability and adherence, one also must consider the cost effectiveness of MPTs. Quaife et al. conducted a cost-effectiveness analysis on MPTs among younger and older women, and female sex workers (FSW) in South Africa [138]. The authors estimated the cost effectiveness of five coformulated or coprovided MPTs (oral PrEP, IVR, injectable ARV, microbicide gel, and SILCS diaphragm used with a vaginal gel) and determined end-user preference by predicting uptake by utilizing a discrete choice experiment approach [136]. In this first MPT cost-effectiveness study, to our knowledge, the authors determined that MPTs would be cost effective mostly among younger women and FSW. The fact that the products served for more than one indication made them more appealing and acceptable to potential users, and therefore, economies of scale from product use and the costs associated with unplanned pregnancies prevented would reduce the net costs of the overall intervention [138]. 

Fundamentally, it is critical to align MPT product development with user preferences. If MPT development is efficacious and widely accepted, the rate of HIV/STI, and unplanned pregnancies can be reduced and provide a solution to the unmet needs in sexual and reproductive health. 

## 7. MPT Development Considerations, Challenges, and Limitations 

Although the discussed MPTs have shown successful formulation development, implementation studies, and user acceptance, there are many considerations and challenges to enhance and advance their development. 

Most MPTs in development involve one or more hormonal contraceptives. Although hormonal contraception is highly effective, it does not fulfill the needs of all women. Hormonal contraceptives can have a myriad of side effects such as irregular bleeding, weight gain, pulmonary embolism, and increased risk for cervical and breast cancer [139,140]. These side effects, along with misconceptions of infertility caused by hormonal contraceptives, are driving forces to their discontinuation [141]. Current commercial nonhormonal contraceptives are often ineffective due to low patient compliance or poor partner cooperation (e.g., condoms and spermicide gels) or require surgery for insertion/removal (e.g., copper intrauterine device). The development of a user accepted nonhormonal contraceptive in an MPT will benefit from a high throughput systemic screening approach to discover new drug candidates. The initiative for MPTs suggests utilizing a DNA barcoding technology for genes critical to reproductive pathways, a high throughput screening method [141]. As more nonhormonal contraceptives are discovered, they must be and able to be combined with antivirals with no drug–drug interactions. Ultimately, fabricating nonhormonal MPTs are necessary to fulfill the needs of all women and expand preventative options. 

Furthermore, there are formulation development challenges that could limit MPT progress. For example, it is extremely difficult to incorporate multiple APIs, especially those with different physiochemical properties, in a single formulation to promote sustained drug release with limited side effects and target PK profiles for all APIs in relevant animal models. Additionally, since the majority of MPTs consist of multiple APIs, it is also important to consider drug–drug interactions (DDIs). Contraceptive hormones, especially progestins, are metabolized by CYP3A4 enzymes and some ARVs are known to induce this enzyme. Studies have shown this could potentially increase hormone metabolism, decreasing drug exposure, and resulting in contraceptive failures [142,143]. It has also been shown that serum concentrations of vaginally delivered contraceptive hormones could be reduced when coadministered with an oral ARV regimen due to these CYP-mediated DDIs [144]. Thus, it is crucial to diligently assess DDIs to ensure each API within the formulation remains unaffected by other drugs and remains in its therapeutic window for the entire release duration. Additionally, IVRs are the most developed MPT formulation and hold the most potential to date; however, it is important to note that there is no universally accepted protocol in place for in vitro release studies, such as in vitro release media, media volume, pH, or drug release under agitation [145]. This makes comparisons between various IVR products and in vivo correlations challenging, which can result in many timely iterations in MPT IVR formulation development. 

Lastly, although many MPTs in preclinical development demonstrate safety and efficacy, financial and technical resources to transition these products to clinical evaluation remain limited. Most preclinical MPTs are developed by academic research institutions or small companies supported by government funding. However, in order to transition into clinical testing, significant capitalization is needed for further animal studies and pharmacology testing and will likely need the financial support of large pharmaceutical companies [146]. This is a significant challenge in MPT development and results in many products being unable to reach clinical stages. Holt et al. suggest the development of a global authoritative committee comprised of multidisciplinary experts to review MPT candidates in preclinical development and guide them to clinical evaluation [146]. The committee would generate a benchmark checklist for each MPT to directly compare products and permit top priority MPT candidates available for interested stakeholders to aid in advancing the product to clinical evaluation [146]. This approach could greatly facilitate the translation of promising MPTs from the lab to the clinic. 

Ultimately, the main MPT development considerations are as follows: (1) development of a long-acting or sustained release delivery system to increase product adherence, which can be elicited by utilizing novel material or API options; (2) limited side effects or adverse events; (3) reversible or removable in the case of an emergency; (4) efficacious for all anticipated indications; (5) cost effective; (6) easily accessible in low- to middle-income countries; (7) discrete and women controlled; (8) accepted by end users; (9) product development and commercialization path for clinical evaluation. 

Taking these considerations into account, the future development of MPTs holds great potential for the prevention of HIV/STIs and unplanned pregnancies. Since the MPT landscape is fairly new, most MPTs in development are in preclinical or early clinical stages and years away from registration. Nevertheless, this leaves significant room for improvement and advances in the formulation development and drug delivery landscape. As MPTs become successful and more established, one can expand this concept to the male population as more male contraceptives are being discovered or incorporate other APIs into the delivery systems to expand upon its indications. 

Moreover, MPT development can only advance in the future as more APIs, materials, manufacturing technologies, and delivery systems are being developed to aid in its fabrication, efficacy, and acceptability. 

## 8. Conclusions 

Daily oral dosage forms for HIV PrEP, STI treatment, and contraception have high failure rates due to low patient adherence. Developing sustained-release MPTs can provide an opportunity to reduce HIV/STI prevalence and unplanned pregnancies in a novel all-in-one platform and achieve high end-user compliance. MPTs in development are formulated in a range of dosage forms (oral, parenteral, and vaginal), and each has its unique highlights and limitations (Table 2).

To our knowledge, there is only one orally delivered MPT in development by the Population Council as a daily dosing regimen, which may not mitigate the lack of user adherence. However, there are promising extended-release oral formulations in development for HIV PrEP and treatment, and if adopted, can have the potential to act as an MPT. Conversely, parenteral formulations can achieve long-acting release, thus reducing the dosing frequency and increasing user adherence. It is important to note that parenteral delivery systems, specifically injectables, are most favored by women in South Africa, where HIV prevalence is highest. However, parenteral formulations should be reversible in case of adverse events, terminate treatment, limit invasive surgery techniques for the insertion/removal of the device, and must consider the therapeutic tail and oral lead-ins. Lastly, the majority of MPTs in development are vaginal dosage forms with IVRs being the most innovative, established, and promoting the sustained release. Nevertheless, there are limitations in IVR design and performance because of their current manufacturing processes. Technological advances in the fabrication process are being established to overcome limitations in drug loading, material, and API choices, and controlled drug release kinetics. These emerging technologies are critical for the efficient and efficacious development of next-generation MPT IVRs. Moreover, implementation studies are crucial for determining successful patient uptake of an MPT since an effective formulation may not be accepted in the user population due to dosing frequency, appearance, comfort, and/or cost. Ultimately, the success of all MPT dosage forms is highly dependent on formulation development for efficacy and implementation studies to determine user preference, acceptability, and adherence (Figure 4). 

## Figures and Tables

**Figure 1 polymers-13-02450-f001:**
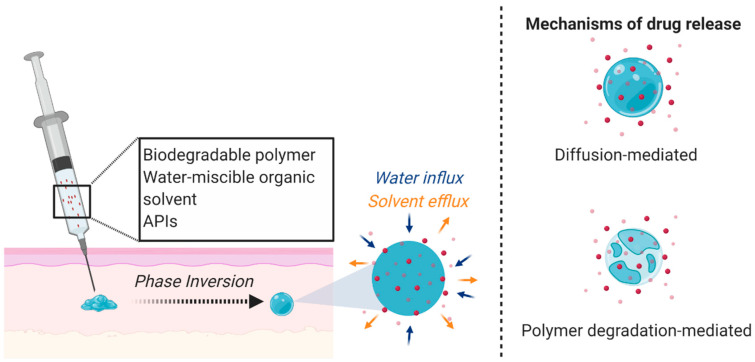
ISFI mechanism and drug release. A liquid solution incorporating a biodegradable polymer, water-miscible organic solvent, and APIs is subcutaneously administered and undergoes a phase inversion due to solvent exchange under physiological conditions to form a semi-solid or solid drug-loaded depot. The APIs are released through diffusion or as the polymer degrades. Figure created with BioRender.com, accessed on 7 October 2020.

**Figure 2 polymers-13-02450-f002:**
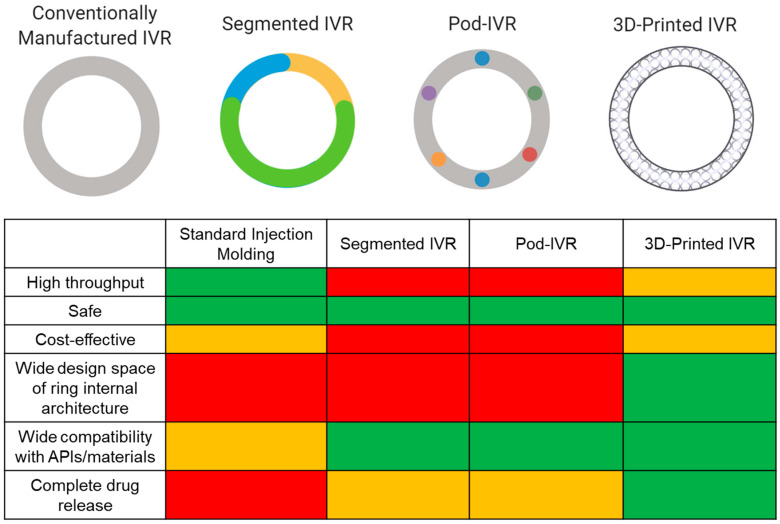
Comparison of IVR technologies. Conventionally manufactured (e.g., injection molding or hot-melt extrusion), segmented, pod, and 3D-printed IVRs. Different colors in the segmented and pod ring design allude to differences in material and APIs, respectively. A relative comparison of IVR characteristics and performance between fabrication methods is shown by the matrix. Green represents the best case, orange is the middle range, and red is the worst case. Figure created with BioRender.com, accessed on 19 July 2021.

**Figure 3 polymers-13-02450-f003:**
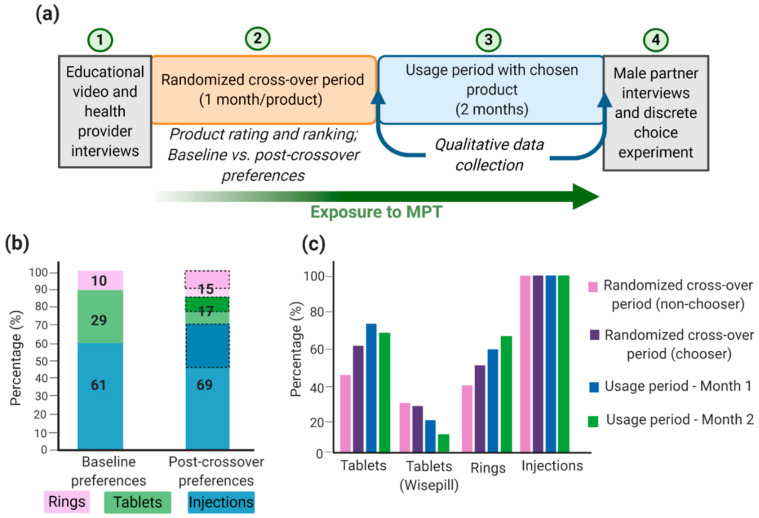
TRIO implementation study design and outcomes: (**a**) TRIO study design; (**b**) women acceptability (%) toward MPTs and shift in preferences from baseline to post-crossover period. Baseline preference is the participant’s choice before trying all products. Post-crossover preference is the participant’s choice after trying all products and the product the woman will use during the two-month usage period. Dashed sections in the post-crossover preferences represent women who had a different preferred product at baseline; (**c**) Product adherence (%) during the study. For tablets, Wisepill containers were used to electronically track the opening of the container and to obtain objective adherence data. Image recreated with permission from [134,136]. Figure created with BioRender.com, accessed on 12 October 2020.

**Figure 4 polymers-13-02450-f004:**
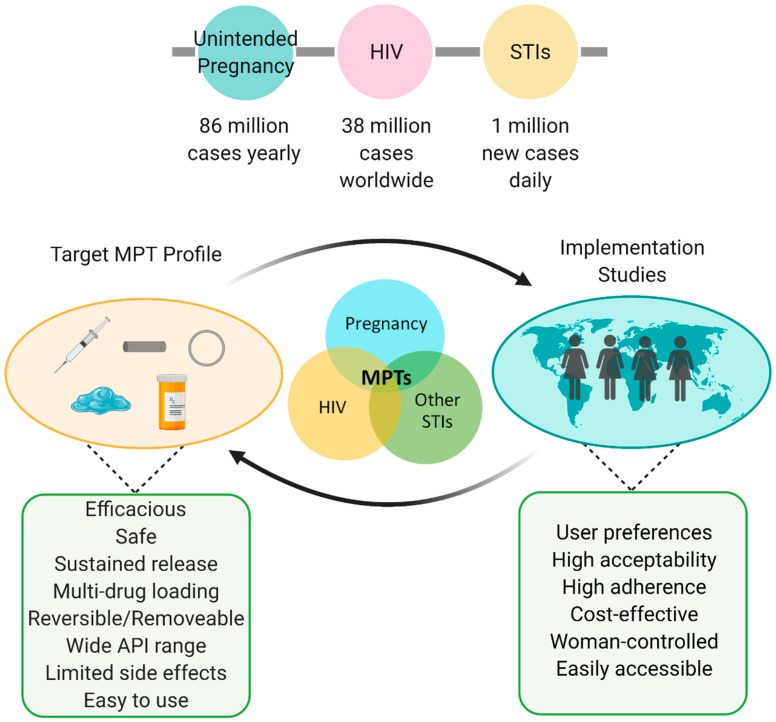
MPT considerations and summary. Globally high prevalence of HIV, STIs, and unplanned pregnancies indicate the urgent need for improved preventative care. Successful MPTs consist of a feedback loop between formulation development (e.g., injectables, implants, intravaginal rings, gels/films, oral tablets) and implementation studies. Figure created with BioRender.com, accessed on 19 July 2021.

**Table 1 polymers-13-02450-t001:** MPTs in clinical development.

Product Name/Developer	Indication	Delivery Platform	Development Stage	Active Pharmaceutical Ingredients	Duration of Action	Reference
International Partnership for Microbicides (originally developed by Karl Malcolm at Belfast University)	HIV, Pregnancy	Intravaginal ring	Clinical—Phase 1	Dapivirine, levonorgestrel	90 days	[34]
Population Council	HIV, Pregnancy	Oral capsule	Clinical—Phase 3	Tenofovir, emtricitabine, levonorgestrel, ethinyl estradiol	24 h	[35]
	HIV, HSV-2, HPV	Vaginal gel	Clinical—Phase 1	carrageenan, MIV-150, zinc acetate	24 h	[36,37]
CONRAD Program (originally developed by Patrick Kiser at University of Utah and Northwestern)	HIV, Pregnancy	Intravaginal ring (Segmented)	Clinical—Phase 1	levonorgestrel, tenofovir	90 days	[38]
CONRAD Program	HIV, HSV-2	Vaginal insert	Clinical—Phase 1	tenofovir, elvitegravir	4–72 h	[39,40,41]
	HIV, HSV-2	Vaginal gel	Clinical—Phase 3	1% tenofovir	12 h	[42,43]
MAPP Biopharmaceutical (originally developed by Deborah Anderson at Boston University)	HIV, HSV-2, Pregnancy	Vaginal film	Clinical—Phase 1	MB66 (monoclonal antibody)	24 h	[44,45]
Evofem Inc.	Chlamydia, Gonorrhea, Pregnancy	Vaginal gel	Clinical—Phase 2	Amphora^®^ gel (L-lactic acid, citric acid, Potassium bitartrate)	Pre-coital	[46,47]
StarPharma	HIV, HSV-2	Vaginal gel	Clinical—Phase 1	SPL7013- VivaGel^™^	24 h	[48,49]

**Table 2 polymers-13-02450-t002:** Comparison of MPT delivery routes and formulations.

Delivery Route	Formulations	Advantages	Limitations	Considerations
Oral	Tablets	Convenient and easy to use Does not interfere with daily activities	Low patient adherence Extended release is difficult to achieve Solubility and absorption challenges Low gastric retention time Significant first-pass metabolism	Mucoadhesive materials Potent APIs Alter delivery architectures to enhance retention time
Parenteral	Implants Injectables	Sustained drug release Compatible with many APIs Injections are most accepted and have the highest adherence by South African women Does not interfere with daily activities Tunable system	Invasive surgery or large needles Implants need to be inserted or removed Injection site reactions Difficulties with reversibility or immediate withdrawal of the drug(s) Typically needs to be administered by a provider	Biodegradable materials Reversible or removable
Vaginal	Films Gels Tablets/Inserts	Convenient and easy to use Facile manufacturing	Extended release is difficult to achieve Difficulties with vaginal bioadhesion May interfere with daily activities Effectiveness is dependent on user adherence	Mucoadhesive materials Slow degrading and non-pH sensitive materials
	Intravaginal rings	Sustained drug release Tunable release kinetics Compatible with many APIs Significant drug uptake and absorption No need for pre-coitus action	Incomplete API release May interfere with intercourse Low user acceptability Limited material and design choices One size/one dose does not fit all Possibility of expulsion	Expand design space Reduce high temperature and pressure systems

## Data Availability

Not applicable.

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
