# Peer review of "Multipurpose Prevention Technologies: Oral, Parenteral, and Vaginal Dosage Forms for Prevention of HIV/STIs and Unplanned Pregnancy"

_polymers, 2021, doi:10.3390/polym13152450_

Round 1

Reviewer 1 Report

A very nice review for this field with good coverages on existing work. I do have some comments:

  1. I do not totally agree with the opinion in Figure 2, particularly for throughput and cost-effectiveness for 3D-printed IVR. So far all 3D-printed IVR requires some level of extrusion with API or CLIP, therefore exposure to high temperature/energy. Furthermore, mass production and low costs are definitely not the most important benefits of 3D printing...
  2. Shall we also highlight the importance of education level for general public? People don't like drugs unless they feel sick. Long term conditions are generally harder to treat in a population with lower education levels.
  3. Please make some level of changes on the figure 4, it almost portrait the MPT has the best solution for this problem. While it may be true in some extend, how about other efforts? 

Author Response

Dear valued reviewer,

Thank you for your helpful feedback to our review article. We have provided a point-by-point response to your comments in the attached file and revised the manuscript to address your comments to the best of our abilities.

Thank you,

Rahima

Reviewer 2 Report

The authors deliver a review manuscript on MPTs deliberating oral, parenteral, and vaginal dosage forms for the prevention of HIV/STIs and unplanned pregnancy.

The authors should discuss the barriers/challenges involved, the effect of route of delivery, and model applications for the development of MPT dosage forms.

The authors discussed two oral, two implants, one patch in this manuscript. The authors mentioned there are recent review articles for vaginal MPTs. What are the criteria for discussing the selective or limited products for each dosage form in the review manuscript?

Author Response

Dear valued reviewer,

Thank you for your helpful feedback and comments. We have provided a point-by-point response to your comments (see attached), and revised the manuscript to address all your comments to the best of our abilities. 

Sincerely,

Rahima
